# 3U CubeSat-Based Hyperspectral Remote Sensing by Offner Imaging Hyperspectrometer with Radially-Fastened Primary Elements

**DOI:** 10.3390/s24092885

**Published:** 2024-04-30

**Authors:** Nikolay Ivliev, Vladimir Podlipnov, Maxim Petrov, Ivan Tkachenko, Maksim Ivanushkin, Sergey Fomchenkov, Maksim Markushin, Roman Skidanov, Yuriy Khanenko, Artem Nikonorov, Nikolay Kazanskiy, Viktor Soifer

**Affiliations:** 1Image Processing Systems Institute NRC “Kurchatov Institute”, Molodogvardeyskaya 151, 443001 Samara, Russia; ivliev@ipsiras.ru (N.I.); podlipnovvv@yandex.ru (V.P.); s.a.fom@mail.ru (S.F.); markushin_max@mail.ru (M.M.); romans@ipsiras.ru (R.S.); artniko@gmail.com (A.N.); 2Samara National Research University Named after S.P. Korolev, Institute of IT and Cybernatics, Moskovskoye Shosse 34, 443086 Samara, Russia; petrov.mv@ssau.ru (M.P.); tkachenko.is@ssau.ru (I.T.); ivanushkin.ma@ssau.ru (M.I.); khanenko.yuv@ssau.ru (Y.K.); soifer@ssau.ru (V.S.)

**Keywords:** hyperspectral images, diffraction grating, NDVI, optical transfer function (OTF)

## Abstract

This paper presents findings from a spaceborne Earth observation experiment utilizing a novel, ultra-compact hyperspectral imaging camera aboard a 3U CubeSat. Leveraging the Offner optical scheme, the camera’s hyperspectrometer captures hyperspectral images of terrestrial regions with a 200 m spatial resolution and 12 nanometer spectral resolution across a 400 to 1000 nanometer wavelength range, covering 150 channels in the visible and near-infrared spectrums. The hyperspectrometer is specifically designed for deployment on a 3U CubeSat nanosatellite platform, featuring a robust all-metal cylindrical body of the hyperspectrometer, and a coaxial arrangement of the optical elements ensures optimal compactness and vibration stability. The performance of the imaging hyperspectrometer was rigorously evaluated through numerical simulations prior to construction. Analysis of hyperspectral data acquired over a year-long orbital operation demonstrates the 3U CubeSat’s ability to produce various vegetation indices, including the normalized difference vegetation index (NDVI). A comparative study with the European Space Agency’s Sentinel-2 L2A data shows a strong agreement at critical points, confirming the 3U CubeSat’s suitability for hyperspectral imaging in the visible and near-infrared spectrums. Notably, the ISOI 3U CubeSat can generate unique index images beyond the reach of Sentinel-2 L2A, underscoring its potential for advancing remote sensing applications.

## 1. Introduction

In recent decades, the space industry has shown growing interest in small-sized space vehicles [1,2,3]. The CubeSat standard, introduced in 1999 [4,5], established a minimal volume unit of 1U, defined as a 10 × 10 × 10 cm cube, with a maximum mass per unit of 1.33 kg [6]. Initially used either as autonomous satellites or as modules of larger apparatuses, early designs included 2U and 3U formats. The standardization allowed carrier rocket manufacturers to develop universal launching systems, independent of CubeSat producers. Subsequently, larger formats such as 6U (30 × 20 × 10 cm), 12U (30 × 20 × 20 cm), and 27U (30 × 30 × 30 cm) were developed, significantly expanding potential applications [5,6,7]. Originally, CubeSats were primarily intended for testing various space technologies, including amateur radio, space observation, cosmic radiation, and studying Earth’s magnetic field. However, with technological miniaturization, recent years have seen CubeSats designed for conducting actual orbital scientific experiments, particularly in Earth’s remote sensing [8].

Currently, a substantial number of 3U CubeSats are equipped with either low [9] or high-resolution optical sensors, delivering Earth images with spatial resolutions up to 3–5 m [10]. However, hyperspectral equipment is less commonly installed on CubeSats due to power limitations and stringent size constraints [11]. Most orbiting satellites with hyperspectral imagers operate in the non-visible shortwave infrared (SWIR) spectral range [11,12,13,14]. For instance, the spectrometer of the commercially designed GHGSat-D spacecraft, operating in the 1600–1700 nm range for greenhouse gas detection, required the construction of a 15 kg 16U CubeSat [11]. Miniaturized hyperspectral cameras find more and more practical applications in aerial and unmanned aerial vehicle (UAV)-based remote sensing and used in fields that are typical for LANDSAT or SENTINEL multispectral images, particularly water monitoring [12,13,14] in different agricultural tasks [15,16,17], forestry [18,19], and other applications [20,21,22,23].

CubeSats also operate in various electromagnetic spectrums, such as infrared (IR) [13], submillimeter band [24], and even gamma-rays [25]. To date, only one hyperspectrometer operating in the partially visible range (290–500 nm) exists on an operating CubeSat, NACHOS, designed by the Los Alamos National Lab for gas analysis with a surface resolution of about 400 m [26]. In 2022, the 3U Aalto-1 hyperspectral nanosatellite completed its mission [27]. Its Aalto-1 Spectral Imager (AaSI) module, a complex Fabry-Pérot Interferometer (FPI) operating in the 500–900 nm range with a spectral resolution of 10–30 nm, is capable of imaging in 6 to 20 channels (out of 60 possible), capturing spatial imagery only in one to three bands simultaneously, with a ground pixel size of 240 × 240 m.

Another 6U CubeSat designed for hyperspectral imaging by the Norwegian University of Science and Technology and the Svalbard Science Centre, intended for ocean color imaging for marine research, employs a grating-based dispersive element and Commercial-Off-The-Shelf (COTS) components in a long optical scheme [28]. It captures images in the visible and near-infrared spectrum (300–1000 nm) with a 5 nm spectral resolution and a ground pixel size of 49 × 60 m. The most advanced hyperspectral imaging on a small satellite (though not a CubeSat) is featured on the GaoFEN-5 satellite designed by the Shanghai Academy of Spaceflight Technology (SAST) [29], which includes two separate spectral units based on the Offner scheme, each optimized for its specific optical range.

These examples demonstrate that the Offner design, adopted in our CubeSat hyperspectrometer, is ideal for nanosatellites due to its efficient use of space without sacrificing technical performance. Our 3U CubeSat spectrometer design offers a spectral range of 400–1000 nm with a 12 nm spectral resolution (150 channels captured simultaneously per pixel) and an effective ground pixel size of 200 × 200 m. This performance surpasses the state-of-the-art hyperspectral capabilities in the visible spectrum of any other 3U CubeSats and even some of the larger 6U designs. Future plans include adding SWIR (1400 to 3000 nm) capabilities to a larger CubeSat platform.

## 2. Earth’s Remote Sensing Platform Based on a 3U Nanosatellite

The SXC3-219 (ISOI) satellite is a collaborative project between researchers at the Image Processing Systems Institute of the Russian Academy of Sciences (IPSI RAS) and the aerospace company JSC ‘SPUTNIX’. This space vehicle (SV) was developed using the 3U OrbiCraft-Pro platform manufactured by SPUTNIX [30,31], and is similar to the SV described in [32].

The SXC3-219 satellite is composed of three distinct units. The first unit houses a reaction wheel-based attitude control system that facilitates the satellite’s orientation in space. The second unit is dedicated to housing the control electronics, while the third unit is exclusively designed to accommodate the payload—specifically, a camera for Earth remote sensing. Gallium arsenide solar cells are mounted on the side surfaces of the units for power generation. A detailed layout of the payload’s integration within the satellite and the external appearance of the satellite are presented in Figure 1a,b.

From Figure 1, it is apparent that the payload lens extends beyond the satellite’s payload unit, requiring additional space in the launch container. Unfortunately, there was not enough time to collaborate with the OrbiCraft team before the launch to optimize the component layout to make extra space for the lens. As a result, we had to decide not to include the onboard high-speed X-band radio transmitter, a standard component of the OrbiCraft platform [30,31]. This decision meant we had to use the low-bandwidth ultra-shortwave (USW) transmitter for data transmission, but it also meant we could finish development and testing in time for launch, providing vital in-orbit test data to guide the planning for the next mission.

The OrbiCraft-Pro’s USW transceiver operates within a 434–436 MHz adjustable frequency range with a bandwidth of less than 20 kHz [30]. It achieves a data transmission rate of 9600 baud in Gaussian minimum shift keying (GMSK) modulation mode, adequate for exchanging telemetry, commands, and modest payload data. The system’s maximum output power of 1 W and sensitivity of −119 dBm are optimized for reliable data transmission from low-Earth orbit.

## 3. Hyperspectral Camera Optical Design

Our hyperspectral camera for remote sensing utilizes an Offner optical system. The hyperspectrometer was originally designed to capture high-resolution (<10 m) images of the Earth’s surface with an angle of view of approximately 1.5°. This design incorporated a catadioptric lens with a 300-mm focal length, as shown in Figure 2a [34]. However, the lens’s size was incompatible with a 3U-format satellite. Consequently, we opted for the compact, commercially available Pentax C2514-M lens [35], which has a 25 mm focal length and an expanded angle of view of approximately 11°. Given the proprietary nature of this lens, Figure 2b represents it in its entirety as a ‘black box’.

In the optical scheme, the Pentax lens captures light and directs it through the slit diaphragm to a spherical mirror. This mirror reflects the light towards a mirror diffraction grating. Upon reflection, the light is decomposed into its spectral components and is then redirected by the spherical mirror to form the final spectral image on the photoreceiver.

### 3.1. Modeling the Image Formation

As noted earlier, the detailed optical scheme of the Pentax C2514-M lens is not publicly available. However, experimental measurements of the lens’s optical transfer function (OTF) have been documented [35]. Figure 3 illustrates the OTF along the optical axis and off-axis at an angle of 9°, with the dashed line representing the meridian plane.

We utilized the published data to construct a theoretical representation, employing a parameterized approximation to emulate the actual Pentax lens within our optical simulation of the imaging system. To distinguish it from the physical Pentax lens, we will refer to our simulation-specific representation as the simplified ‘optical approximation’ of the lens.

Figure 4 indicates that at high frequencies, the OTFs for off-axis beams 2 and 3 closely align. However, at low frequencies, there is a notable difference in the OTF for off-axis incidences. Of particular importance is the region near 100 mm^−1^ because it is near the frequency of the sensor elements used (133 mm^−1^).

Figure 5 illustrates the on-axis point-spread function (PSF) of our optical approximation of the lens as a 3D pattern (Figure 5a) and its *x*-axis profile (Figure 5b).

Parameters of our optical approximation of the lens were inputted into the simulated optical design in place of the actual Pentax C2514-M parameters. Utilizing ZEMAX 18.4.1 software, the overall OTF of the hyperspectrometer was calculated. Figure 6 depicts the resulting OTF plot.

The fluctuations in the OTF observed in Figure 6 may be attributed to the process of light propagation through the slit diaphragm of the spectral block. The results from the numerical simulation of the hyperspectrometer’s performance while in the orbital motion are discussed in refs. [35,36,37,38].

### 3.2. Hyperspectrometer Design and Fabrication Technique

#### 3.2.1. Photometric Calculations and Selecting a Photosensitive Array and Calculator

A typical optical setup for capturing a hyperspectral image of an object on a photosensitive array is shown in Figure 7. An object with a diffuse reflection coefficient ρ(λ) is illuminated by a light source with spectral luminance intensity E_λ_(λ). The light reflected from the object traverses the atmosphere with transmission coefficient τ_1_(λ) and is then collected by the lens system, characterized by a focal length *f*, input pupil diameter D, and transmission coefficient τ_2_(λ). It then passes through a hyperspectrometer spectral grating-like unit—represented here as a prism for its intended optical functionality—with transmission coefficient τ_3_(λ) and the dispersion element’s diffraction efficiency (DE) ε(λ). The photosensitive array, with a quantum efficiency (QE) of δ(λ), is positioned at the hyperspectrometer’s active area (image plane) to capture the partial image formed by the lens, which is then decomposed by the dispersion element into N spectral columns on the photosensitive array, resulting in a spectral irradiance *E*_λ1_(λ).

The relationship between the spectral irradiances in the object plane and in the image plane, when *f* << *H*, is described by a standard photometric formula for illuminance on the imaging plane for a lens with a focal ratio *k* = *f/D*, considering given losses:(1)$$E_{\lambda 1}\left(\lambda \right)=\frac{\tau_{1}\left(\lambda \right)\tau_{2}\left(\lambda \right)\tau_{3}\left(\lambda \right)\varepsilon \left(\lambda \right)\delta \left(\lambda \right)\rho \left(\lambda \right)}{4k^{2}}E_{\lambda }\left(\lambda \right),$$

The irradiance per pixel is calculated by integrating over a spectral range:(2)$$E_{1}=\int_{\lambda_{1}}^{\lambda_{2}}\frac{\tau_{1}\left(\lambda \right)\tau_{2}\left(\lambda \right)\tau_{3}\left(\lambda \right)\varepsilon \left(\lambda \right)\rho \left(\lambda \right)E_{\lambda }\left(\lambda \right)}{4k^{2}}d\left(\lambda \right),$$
where λ_1_, λ_2_ are the limits of the spectral range detectable by the sensor. Given the hyperspectrometer’s narrow spectral range, an approximate equation may be used in place of (2), which still provides high accuracy:(3)$$E_{1}\approx \frac{\tau_{1}\left(\lambda \right)\tau_{2}\left(\lambda \right)\tau_{3}\left(\lambda \right)\varepsilon \left(\lambda \right)\rho \left(\lambda \right)}{4k^{2}}E_{\lambda }\left(\lambda \right)∆\lambda ,$$
where Δλ is the bandwidth of a hyperspectrometer channel. It is important to note that the spectral irradiance of the object is inferred from:(4)$$E_{\lambda }\left(\lambda \right)=\tau_{4}\left(\lambda \right)X\left(\lambda ,T\right),$$
where τ_4_(λ) is the atmospheric transmission coefficient, differing from τ_1_(λ) because sunlight traverses the atmosphere at varying angles before reaching the Earth’s surface, thus necessitating the use of both coefficients. *X(λ,T)* represents the spectral radiance function of blackbody (BB) radiation. In Earth remote sensing (ERS), the sun, with T = 5780 K, acts as the light source.
(5)$$X\left(\lambda ,T\right)=\frac{2\pi c^{2}h}{\lambda^{5}}\frac{1}{exp\left(\frac{hc}{\lambda kT}\right)-1}.$$

Figure 8 displays a plot of the spectral density of radiation versus wavelength, calculated using Equation (5).

It is important to note that the spectrum of light actually reaching the Erath’s surface is somewhat altered, featuring several absorption bands of main atmospheric gases [11,32]. Moreover, as light traverses the atmosphere twice, this results in enhanced absorption.

The quantity of light quanta per unit time, Φ (flux), striking a single array pixel is expressed by:(6)$$\Phi =\frac{{{\left(∆x\right)}^{2}\tau }_{1}\left(\lambda \right)\tau_{2}\left(\lambda \right)\tau_{3}\left(\lambda \right)\varepsilon \left(\lambda \right)\rho \left(\lambda \right)E_{\lambda }\left(\lambda \right)∆\lambda \lambda }{4k^{2}hc},$$
where Δx is the array pixel size.

The number of signal electrons, *n*, generated in a pixel due to the flux Φ during the accumulation time *t* is:(7)$$n=\delta \left(\lambda \right)\Phi t=\frac{{{\left(∆x\right)}^{2}\tau }_{1}\left(\lambda \right)\tau_{2}\left(\lambda \right)\tau_{3}\left(\lambda \right)\varepsilon \left(\lambda \right)\delta \left(\lambda \right)\rho \left(\lambda \right)E_{\lambda }\left(\lambda \right)∆\lambda \lambda t}{4k^{2}hc}.$$

The overall noise level from photovoltaic conversion comprises both the photon noise of the object radiation flux and the inherent noise of the charge coupled device (CCD)-array sensor and electron duct. Since photon noise is challenging to assess, our calculations will consider only the sensor noise, namely, the number of thermal electrons, *n_t_* (a parameter specified by the CCD-camera manufacturer).

For an acceptable quality hyperspectral image, a practical requirement is for the number of signal electrons to be at least double that of the thermal electrons at the spectral band edges. The accumulation time, *t*, is the sole variable parameter during image capture:(8)$$t=\frac{8n_{t}k^{2}hc}{{{\left(∆x\right)}^{2}\tau }_{1}\left(\lambda \right)\tau_{2}\left(\lambda \right)\tau_{3}\left(\lambda \right)\varepsilon \left(\lambda \right)\delta \left(\lambda \right)\rho \left(\lambda \right)E_{\lambda }\left(\lambda \right)∆\lambda \lambda }.$$

This accumulation time is a critical factor influencing the resolution of a hyperspectral image along the satellite’s path. In scanning mode, the on-path resolution, Δ*Y*, is determined by the accumulation time:(9)∆Y=v·t,
where *v* is the satellite projection’s velocity over the Earth’s surface. Although the resolution could be enhanced by using a pitch mode, this feature is not available on our satellite. The on-surface resolution across the satellite path, ΔX, depends on the photosensitive array pixel size and the lens’s focal length:(10)$$∆X=\frac{h\cdot ∆x}{f},$$
where *h* is the orbital altitude. Improving resolution by reducing pixel size, Δx, would inversely affect the on-path resolution Δ*Y*. In our calculations, we chose a CCD-array based on the condition that Δ*X* and Δ*Y* are approximately equal. This criterion is met by complementary metal oxide semiconductor (CMOS)-arrays with 3–4 µm pixels, such as Sony IMX296.

The accumulation time for a hyperspectrometer equipped with a Pentax C2514-M lens (k = 1.4) and an IMX296 CMOS-array [39] (pixel size Δx = 3.45 µm, thermal noise n_t_ = 41, and maximum product of quantum efficiency and diffraction efficiency ε × δ_max_ = 0.02) with N = 150 spectral channels can be estimated. For simplification in the evaluation, we made the following assumptions:The atmospheric transmission function τ_4_(λ) is assumed to be equivalent to τ_1_(λ) and approximately 0.8 in the wavelength range of 400–1000 nm.The Earth’s diffuse reflection coefficient ρ(λ) is assumed to be roughly 0.5.The transmission coefficients for the lens and hyperspectrometer optical systems are considered constant, with τ_2_(λ) = τ_3_(λ) ≈ 0.8.

This evaluation process will consider the spectral functions ε(λ) and δ(λ), which vary significantly across the wavelength range of 400–1000 nm. For example, despite an optimal microrelief height of the diffraction grating ensuring equal diffraction efficiency (DE) at both operating waveband extremes, the DE value fluctuates between 0.6 and 1, as shown in Figure 9. Similarly, the relative response (RR) of the photosensor demonstrates considerable variation, as depicted in Table 1 [39].

Table 1 illustrates that the relative response (RR) of the sensor declines sharply after reaching a peak as the wavelength increases. Therefore, to maintain a consistent ε(λ)δ(λ) across the spectrum, the diffraction grating’s microrelief height should be adjusted. An increased height of 180 nm is proposed to ensure that the ε(λ)δ(λ) remains approximately constant at both ends of the operating waveband. Figure 10 displays the DE binary diffraction grating with this adjusted microrelief height, showing variation from 0.06 at the wavelength edges to a maximum of 1 at 720 nm.

Substituting the numerical values into (8) yields an estimated accumulation time *t* of approximately 0.016 s for the daylight conditions on a sunny day. With the satellite moving at a velocity of 7.66 km/s, taking into account the speed of the subsatellite point, we have a distance of approximately 125 m. In that case, the cross-path resolution ΔX is approximately 200 m and the on-path resolution ΔY is estimated to be approximately 322 m.

#### 3.2.2. Layout and Mounting of the Optical Elements

To mitigate the impact of thermal deformations on image quality, the optical elements were radially mounted within the spectral unit. Reference [35] demonstrates that this configuration results in negligible variation in the polychromatic point-spread function (PSF) across a temperature range of −40 °C to 45 °C and a wavelength range of 400 nm to 1000 nm. Figure 11 illustrates a 3D model of the hyperspectrometer design.

The lens (not shown in Figure 11) is attached to the hyperspectrometer using adapter (1), with the slit diaphragm (2) secured at its base by a thread ring. The adapter and photosensor (10) are affixed to the cap (3) with screws, which in turn is connected to the case (7). Inside the case, the diffraction grating (6) and the spherical mirror (8) are coaxially mounted using their respective holders (5 and 9). The grating holder (5) can be moved by rotating the adjustment ring (4), and the mirror holder (9) can be adjusted using a thread coupling for optical system alignment. Both the mirror and the grating are fixated using thread rings, facilitating focus adjustment by moving these components along the optical axis.

The components, such as the cases, holders, and screw rings, are manufactured from a hard aluminum alloy with a low thermal expansion coefficient. The finalized design weighs no more than 1.6 kg and has dimensions of 146 mm × 94 mm × 94 mm (length × width × height).

#### 3.2.3. Technique for Synthesizing Dispersive Optical Elements on a Convex Surface

The fabrication of the dispersive optical elements utilized a technological process outlined in reference [40]. In the proposed optical system, a diffractive element synthesized on a convex surface (mirror grating) is crucial for the quality and quantity of information processed [41]. The traditional fabrication method, involving successive stages of metal layer deposition, mask patterning, plasma chemical etching, and mirror layer application, is not suitable for high-quality elements due to increased roughness in the etched unmasked areas, which leads to significant optical losses.

After experimental studies, we proposed eliminating the etching stage from the fabrication process and instead adopting a different technique to synthesize the grating mask. The revised procedure for fabricating a mirror diffraction grating on a convex surface is as follows:Stage 1. The surface of the UV grade fused silica (KU-1) substrate with a refractive index n = 1.46 (for 532 nm) is cleansed of contaminants using a surface-active substance, then rinsed with distilled water, and dried with compressed air to remove residual moisture.Stage 2. A chromium (Cr) film is deposited onto the cleaned substrate surface. This deposition is carried out in an automated magnetron sputtering system, Caroline D12A. The substrate is placed into the system’s vacuum chamber, where air is evacuated to a residual pressure of 1.5 × 10^−3^ Pa. The substrate is heated to 120 °C, and the deposition is performed by sputtering a chromium target of 99.99% purity in an argon atmosphere at an argon consumption rate 2 l/h and a pressure of 1.5 × 10^−1^ Pa. The target thickness for the film is 180 nm, with a deposition time of approximately 9 min. After the deposition, the substrate is left in the chamber to cool to room temperature before removal.Stage 3. The diffraction grating pattern is written into the chromium film on a circular laser writing station, CLWS-200 [40]. The substrate, now coated with the 180 nm chromium layer, is secured on a rotating holder and centered. A digital file detailing the desired grating structure is input into the CLWS control system. The grating is inscribed using 532 nm in polar coordinates, modulating the rotation speed and the linear position to form a protective oxide film over the chromium layer in the laser-exposed regions. Due to the curved substrate surface, the laser is automatically focused in this stage, as well.Stage 4. Post-inscription, an oxide mask is present on the chromium film, facilitating the selective removal of chromium in unmasked areas using a special liquid developer, while the areas under the oxide film are preserved. This development takes roughly 5 min. Following this, the deposition process from Stage 2 is repeated, effectively eliminating the need to transfer the microrelief pattern into the substrate due to the adequate height of the chromium film, presumed to be a quarter-wave height for visible light.Stage 5. The depth of the diffraction grating beneath the mask is verified with a contact profiler, specifically the KLA Tencor P16+.Stage 6. The mirror layer deposition onto the substrate surface with the 180 nm structure is carried out through successive application of chromium and aluminum films to the required thickness, utilizing the Caroline D12A magnetron deposition system. The substrate is positioned within the vacuum chamber of the system, where the air is pumped out to achieve a residual pressure of 1.5 × 10^−3^ Pa. During the heating phase, the substrate is brought to a temperature of 120 °C. Chromium sputtering from a 99.99% pure target proceeds in an argon environment, with an argon consumption of 2 l/h at a pressure of 1.5 × 10^−1^ Pa, achieving a film thickness of 25 nm within approximately 2 min. The chromium film, which has a high level of adhesion to the quartz surface, forms the base layer for the reflective surface. Aluminum from a 99.99% pure target is then sputtered under the same conditions to a thickness of 125 nm, taking around 12 min. After the deposition, the substrate is cooled to room temperature before removal.Stage 7. The height of the produced mirror element is verified using a KLA Tencor P16+ surface profiler. The successful implementation of this new methodology is visually confirmed by a profilogram of the mirror grating, indicative of potential image quality enhancements. Figure 12a,b exhibit the profilograms of diffraction gratings created by the conventional plasma chemical etching and the new sputtering method, respectively.

The proposed technique’s advantage lies in its reduced number of fabrication steps. Moreover, the synthesized optical element exhibits higher quality due to the implementation of direct laser writing and the avoidance of direct mechanical contact with the substrate surface. A ‘lift-off’ process was employed to create the diffractive slit. A slit measuring 10 μm in width and 16 mm in length was produced by applying a photoresist FP-9120 onto a quartz substrate via centrifugation at 3000 rpm, followed by drying.

Next, a photomask was precisely aligned, and the photoresist was exposed to UV light, resulting in a 10 µm wide and 16 mm long strip after submersion in a dilute NaOH solution. This process revealed the substrate area and maintained the photoresist film above the intended diffractive slit. A 120 nm thick chromium film was then thermally sputtered onto it. Chromium’s strong adhesion to the resist-free areas ensures a firm bond to the quartz substrate, while it simply coats the photoresist surface in the slit area without bonding to the quartz. After dissolving the photoresist in acetone, the chromium film overlay was removed leaving behind a precise 10 µm by 16 mm diffraction slit in the chromium film. The fabrication of the slit is a technically intricate process, schematically depicted in Figure 13.

## 4. Remote Sensing Experiment

### 4.1. Generating a Hypercube

The space mission commenced on 9 August 2022, with the launch of the Soyuz-2.1b spacecraft with a Fregat booster from the Baikonur Cosmodrome. The satellite was placed into orbit at an altitude of 450 km.

The initial target for hyperspectral imaging was deliberately selected to include a water body within the frame, which facilitates straightforward comparison with Sentinel-2 L2A imagery, the latter serving as a benchmark. The image area is delineated by a contour in Figure 14. Given the USW channel’s limited bandwidth, we restricted the hyperspectrometer range to 350 central pixels. This limitation allowed for the transmission of a single hyperspectral frame over 50–60 communication sessions over 8 days.

The data captured by the hyperspectrometer was assembled onboard the spacecraft into a video stream, with each frame representing the spectral distribution from one column of the hyperspectral image. This stream was decoded using PyPylon Python library, which interfaces with Basler pylon 1.3 camera software. The imaging parameters were defined by a JSON configuration file transmitted to the satellite from the ground station. Next, the video data were processed to construct the hyperspectral image layers.

### 4.2. Thematic Processing of Hyperspectral Data

The acquired hyperspectral image was processed to create a color composite using the 500 nm, 550 nm, and 600 nm channels (shown in Figure 15, left) and was then oriented correctly. For a comparison, a color composite of the same area from Sentinel-2 L2A, captured around the same time, is displayed in Figure 15 (right). Notably, the strip image from the ISOI satellite was nearly perpendicular to that from Sentinel-2 L2A. The latter’s narrower width compared to the length of the former resulted in a shorter comparative strip on the right side of Figure 15.

In the subsequent analysis, the images are placed horizontally, without accounting for the orbital tilt. To validate the hyperspectral images obtained, we processed several index images. Given that Sentinel-2 L2A has a limited number of spectral channels, comparing the normalized difference vegetation index (NDVI) is the most feasible. Figure 16a displays the NDVI distribution over the upper section of the area shown in Figure 15, for which Sentinel-2 L2A NDVI data are available (shown in Figure 16b). The central wavelengths of the channels used to calculate the indices are described in [41].

The NDVI was calculated using the following formula [42]:(11)$$NDVI=\frac{B7-B4}{B7+B4}.$$
where B4 is the range with a central wavelength of 665 nm, and B7 is the range with a central wavelength of 783 nm for Sentinel-2 L2A. For SXC3-219, B4 is the sum of the brightness in the range from 650 to 680 nm, and B7 is the sum of the brightness in the range from 785 to 900 nm (the classical NDVI formula [21]).

Data from our satellite were processed using channels that correspond to those of Sentinel-2 L2A and were displayed in 256 levels of grey for ease of analysis. Meanwhile, Sentinel-2 L2A data were represented in standard pseudo-colors.

Unfortunately, the pseudo-color visualization parameters used by Sentinel-2 L2A are not available to us, which complicates the task of replicating the NDVI visualization in the same format. Additionally, the SXC3-219 (ISOI) satellite lacks precise geolocation capabilities, making a precise area comparison challenging. Therefore, we restricted our comparison to several distinct points within the image (see Figure 17).

Table 2 lists the NDVI values at these selected points, derived from data obtained from SXC3-219 (ISOI) and Sentinel-2 L2A on 10 June 2023.

Table 2 indicates that the NDVI values derived from SXC3-219 (ISOI) and Sentinel-2 L2A data differ slightly. This discrepancy is largely attributable to the significant difference in pixel area between the two satellites; a single pixel from SXC3-219 (ISOI) covers a much larger surface area that one from Sentinel-2 L2A. At points 1 and 3 in Figure 18, the NDVI from SXC3-219 (ISOI) is lower than that from Sentinel-2 L2A data, likely due to the averaging effect over a larger pixel area. Conversely, at point 2, the NDVI is higher for SXC3-219 (ISOI), which also results from averaging over a larger pixel area.

Due to the limited spectral channels of Sentinel-2 L2A, it is not possible to compare additional index images. However, we present two popular indices, the water index (WI) in Figure 18a and the anthocyanin reflectance index (ARI) for vegetation in Figure 18b. These indices are calculated using the following formulas [9]:(12)$$WI=\frac{I_{900}}{I_{970}},$$
(13)$$ARI=0.5+\left(\frac{1}{I_{550}}-\frac{1}{I_{700}}\right),$$
where *I*_550_, *I*_700_*, I*_750_, *I*_900_, and *I*_970_ represent the intensity values in the corresponding spectral channels.

As expected, the WI shows higher values along the water banks, while the ARI distribution appears more localized. The 150 spectral channels of the SXC3-219 (ISOI) satellite facilitate the calculation of numerous index images, but presenting them here is not practical without the capability for ground data comparison.

To assess the performance of the hyperspectrometer, several hyperspectral images were captured over desert regions featuring irrigated circular agricultural fields in northern Saudi Arabia. Figure 19a presents a color-composite image derived from SXC3-219 (ISOI) satellite data, while Figure 19b provides a color-composite image from Sentinel-2 L2A of a nearly identical location. Figure 19c illustrates the NDVI distribution obtained from the SXC3-219 (ISOI) hyperspectral image.

In Figure 19, the NDVI distribution clearly correlates with the location of irrigated fields, showing high NDVI values within these areas, while areas outside exhibit lower NDVI values consistent with vegetation-free desert land.

Upon examining an enlarged section of the image in Figure 20, it is evident that the edges of terrain features are significantly sharper in the direction transverse to the satellite’s path than in the along-path direction. This observation aligns with the actual resolution ratio of 200 m cross-path and approximately 800 m along-path. The hyperspectrometer’s optical system achieves a resolution that closely matches the intended design specifications.

## 5. Conclusions

Despite the formidable size and weight constraints of the compact 3U CubeSat format, our team has successfully engineered and deployed an innovative hyperspectrometer into orbit for Earth remote sensing applications, such as forest monitoring in the visible to near-infrared spectrum (VNIR). While the hyperspectrometer onboard SXC3-219 (ISOI) does not match the superior spatial resolution of Sentinel-2 L2A (200 m compared to 10 m, respectively) and has a more limited wavelength range, it provides a significantly higher spectral channel density within the 400 nm to 1000 nm range, offering 150 channels in contrast to Sentinel-2 L2A’s 9. This enhancement in spectral capacity is especially useful for detailed vegetation analysis, facilitating the use of numerous spectral indexes that are impossible to derive with Sentinel-2 L2A data due to its limited spectral capabilities.

Looking forward, we plan to transition to a larger 6U satellite platform, which will accommodate a high-speed X-band transmitter and a hyperspectrometer outfitted with a 300 mm long-focus lens. This advancement is expected to yield a surface resolution of 7–10 m, thereby broadening the satellite’s utility for a wider array of applied remote sensing challenges.

## Figures and Tables

**Figure 1 sensors-24-02885-f001:**
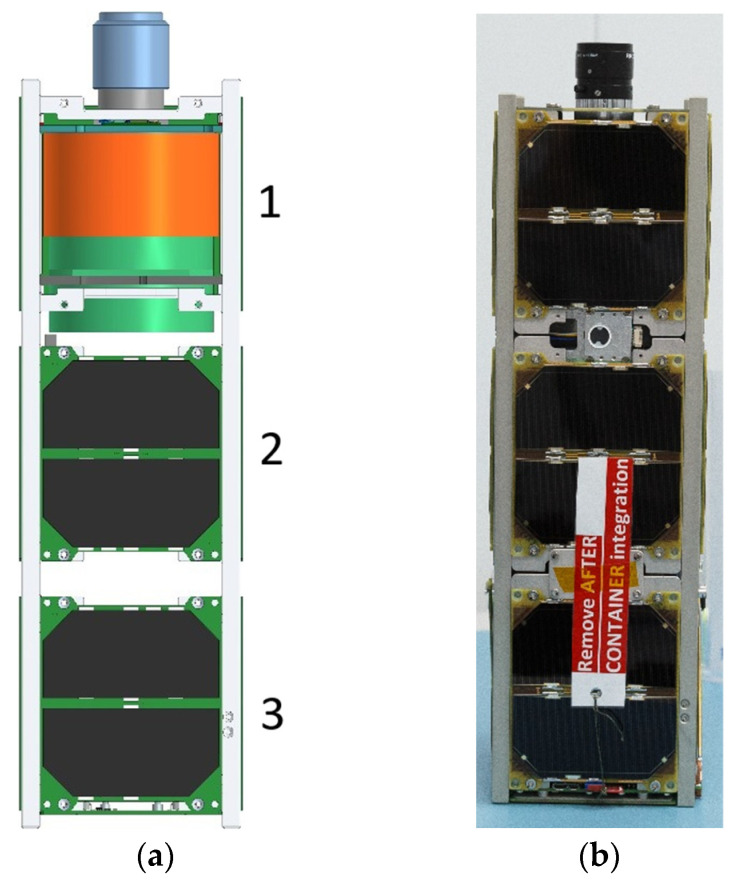
(**a**) Layout of the payload within the CubeSat; (**b**) External appearance of SXC3-219 (ISOI), 1—Offner imaging hyperspectrometer, 2—power supply and processing unit, 3—communication and orientation unit (the satellite structure is presented in detail in [33]).

**Figure 2 sensors-24-02885-f002:**
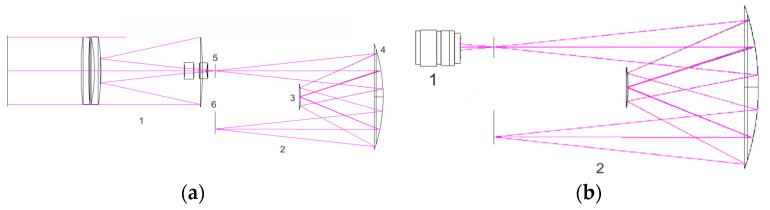
Optical layout of the hyperspectral remote sensing camera: (**a**) optical scheme of the imaging hyperspectrometer featuring a catadioptric lens system (1—lens, 2—spectral block, 3—diffraction grating, 4—main mirror, 5—slit diaphragm, 6—registration plane); (**b**) optical scheme representation of the hyperspectrometer with the Pentax C2514-M lens.

**Figure 3 sensors-24-02885-f003:**
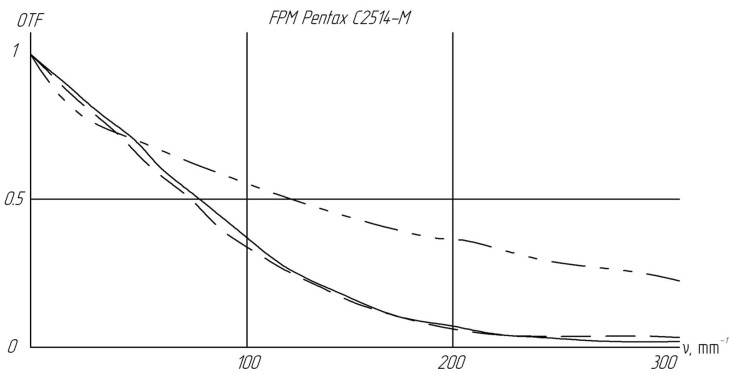
Optical transfer function (OTF) of the C2514-M lens: on the optical axis (solid line), and at a 9° angle off-axis, with the horizontal coordinate indicated by a dashed line and the vertical coordinate by a dotted line.

**Figure 4 sensors-24-02885-f004:**
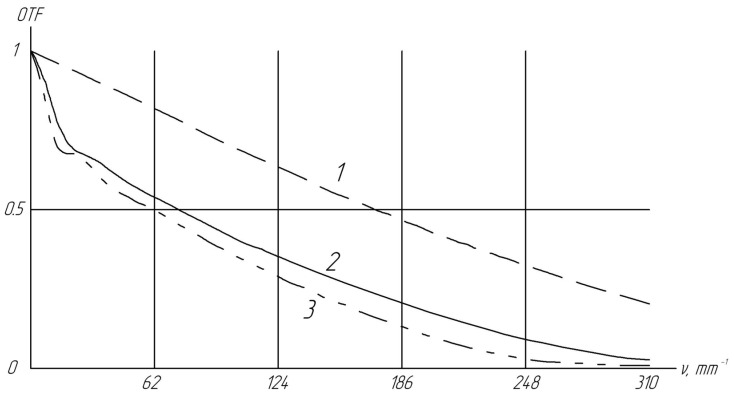
Optical transfer function (OTF) of our Pentax lens optical approximation used for subsequent modeling: 1—OTF for the on-axis direction, 2—OTF for the off-axis direction with a 9° tilt to the *x*-axis, and 3—OTF for the off-axis direction with a 9° tilt to the *y*-axis. Explanation for different lines are shown in Figure 3

**Figure 5 sensors-24-02885-f005:**
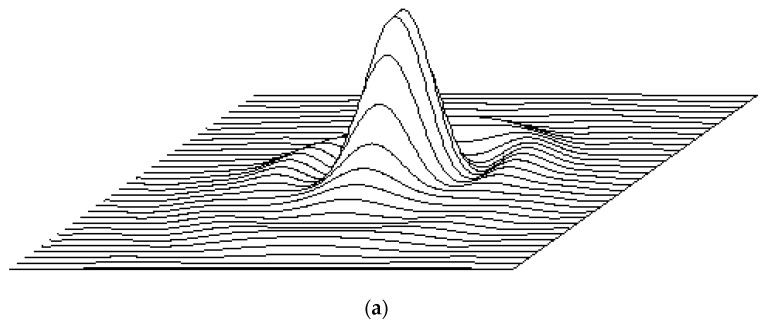
Model of the lens: (**a**) 3D representation of the on-axis point-spread function (PSF), (**b**) the PSF profile along the *x*-axis, and (**c**) the PSF profile along the *y*-axis.

**Figure 6 sensors-24-02885-f006:**
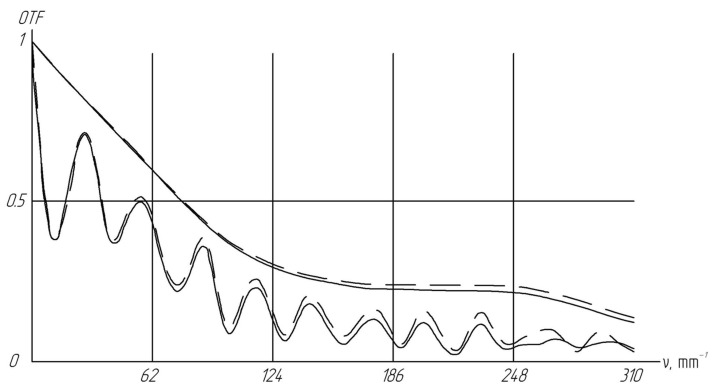
Optical transfer function (OTF) of a hyperspectrometer utilizing an approximated model of the lens with characteristics similar to the Pentax C2514-M lens. Explanation for different lines are shown in Figure 3

**Figure 7 sensors-24-02885-f007:**
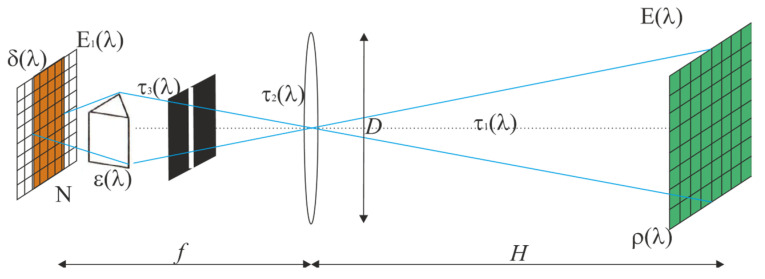
Imaging hyperspectrometer, annotated for photometric calculations.

**Figure 8 sensors-24-02885-f008:**
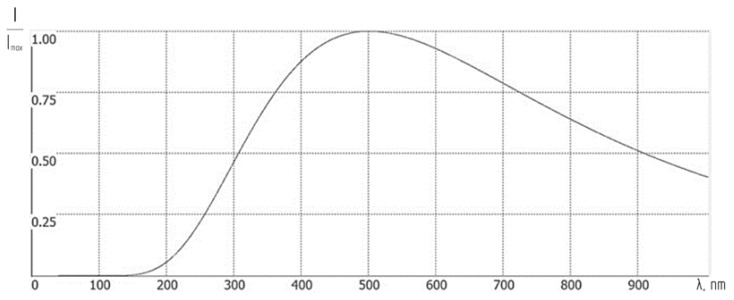
Spectral density of solar radiation versus wavelength, calculated without considering the Earth’s atmosphere influence.

**Figure 9 sensors-24-02885-f009:**
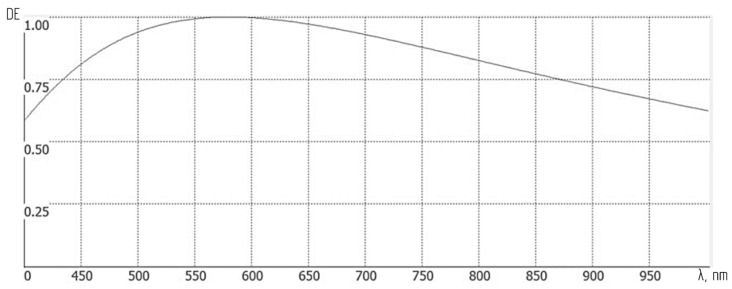
Diffraction efficiency (DE) of a binary reflecting diffraction grating with a microrelief height of 145 nm versus wavelength.

**Figure 10 sensors-24-02885-f010:**
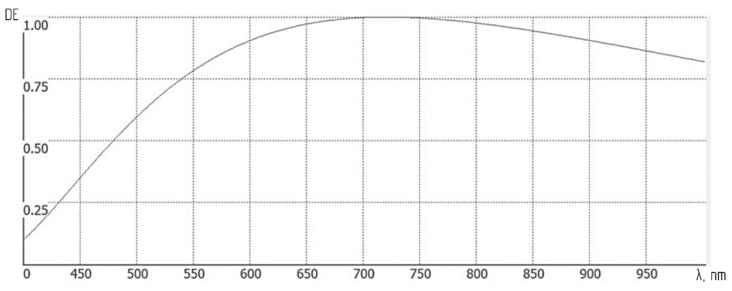
Diffraction efficiency (DE) plot of a binary diffraction grating featuring a microrelief height of 180 nm.

**Figure 11 sensors-24-02885-f011:**
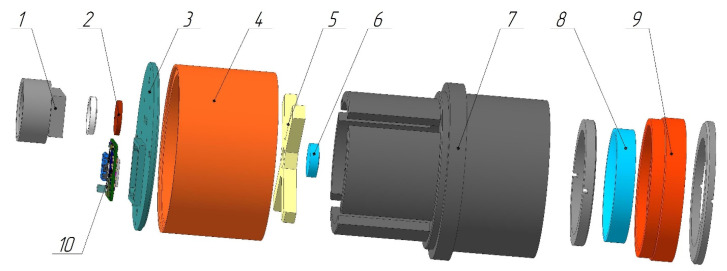
Three-dimensional model of the hyperspectrometer, including: 1—lens adapter, 2—slit diaphragm, 3—cap, 4—adjustment ring, 5—reflecting diffraction grating holder, 6—reflecting diffraction grating, 7—case, 8—spherical mirror, 9—spherical mirror holder, and 10—photosensor.

**Figure 12 sensors-24-02885-f012:**
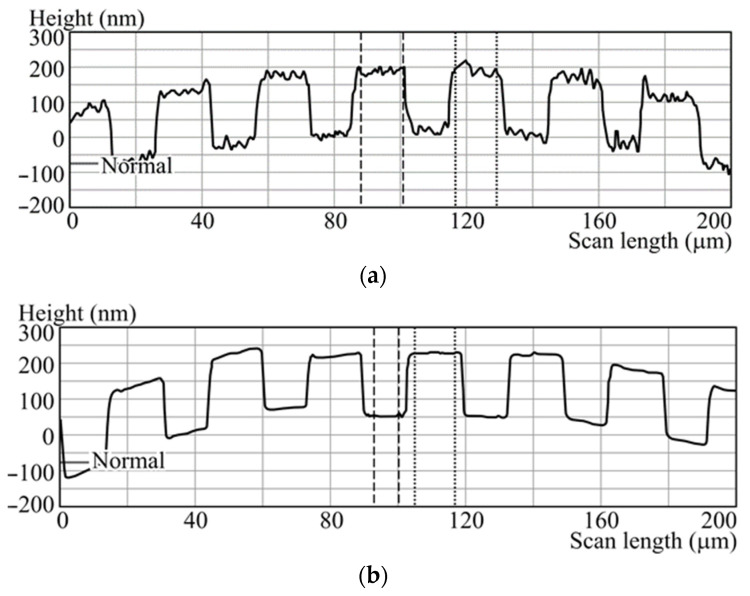
Profilograms of the diffraction grating created using (**a**) plasma chemical etching and (**b**) deposition of a thick mask. Data obtained using a KLA Tencor P16+ profilometer.

**Figure 13 sensors-24-02885-f013:**
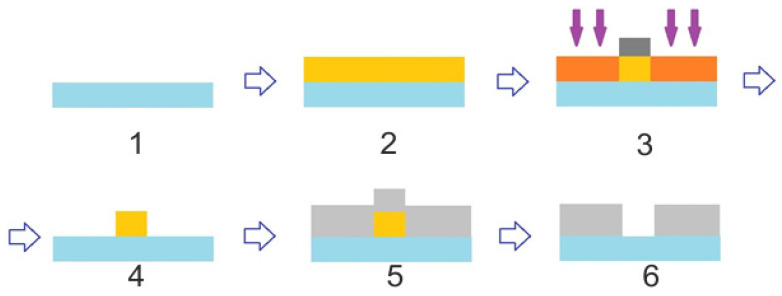
Diagram illustrating the diffraction slit fabrication process, with the following components: 1—fused silica substrate, 2—photoresist deposition, 3—UV exposure through a mask, 4—photoresist development, 5—chrome sputtering, 6—removal of the photoresist and excess chromium.

**Figure 14 sensors-24-02885-f014:**
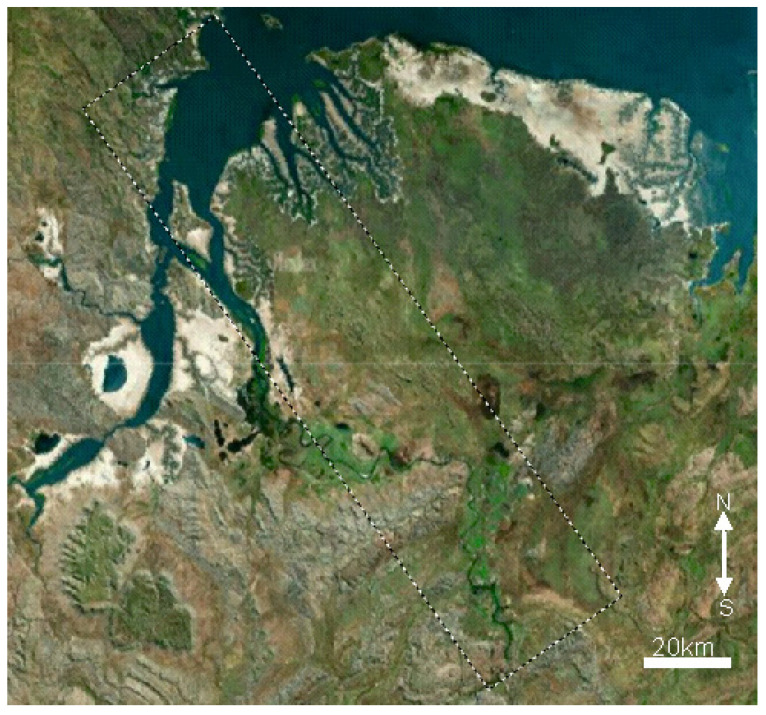
Region targeted for imaging (North Australia) (image from the Yandex Maps service 3 April 2023, -coordinates; 14.843, 128.293).

**Figure 15 sensors-24-02885-f015:**
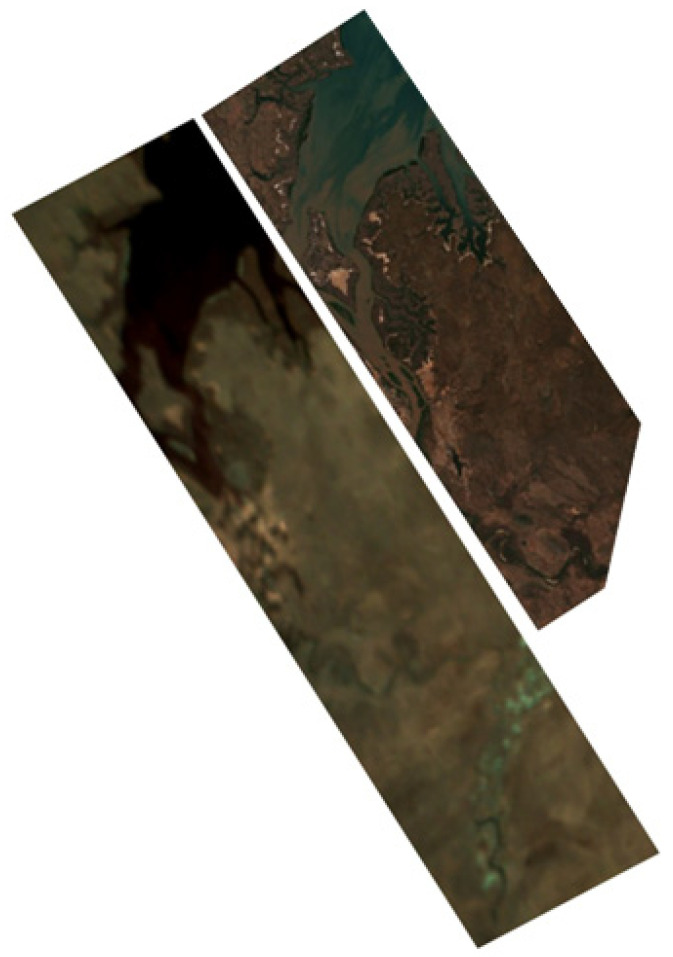
Color-composite images: from SXC3-219 (ISOI) satellite (**left**) and from Sentinel-2 L2A (**right**).

**Figure 16 sensors-24-02885-f016:**
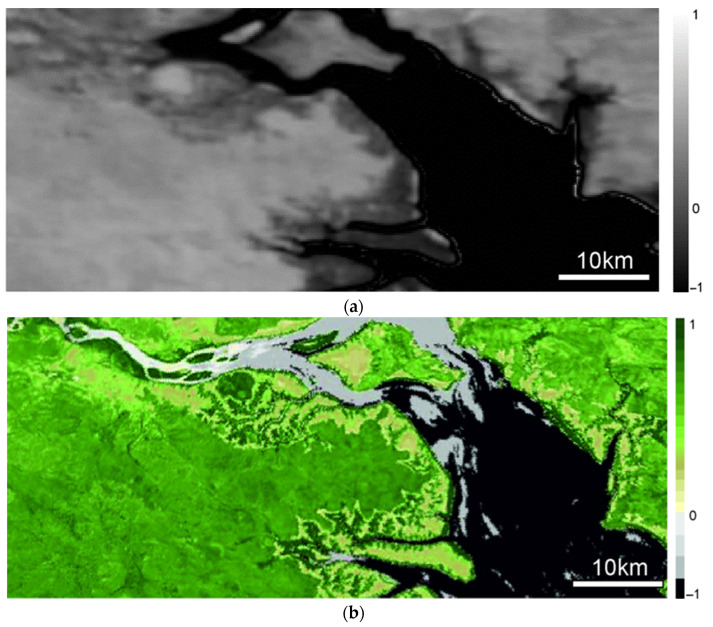
Patterns of normalized difference vegetation index (NDVI) distribution obtained from (**a**) SXC3-219 (ISOI) satellite and (**b**) Sentinel-2 L2A.

**Figure 17 sensors-24-02885-f017:**
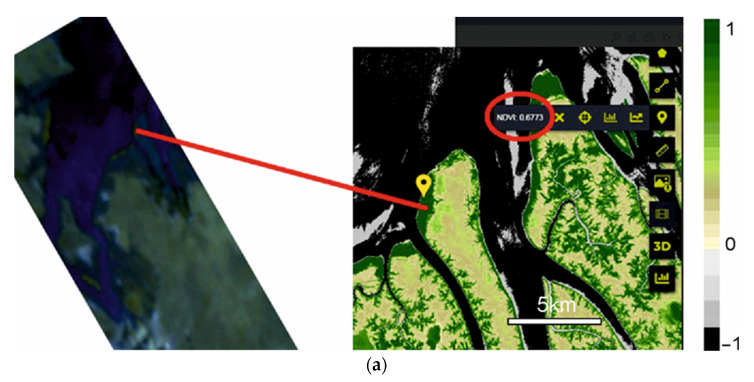
Selected points for comparison: (**a**) point 1, (**b**) point 2, and (**c**) point 3.

**Figure 18 sensors-24-02885-f018:**
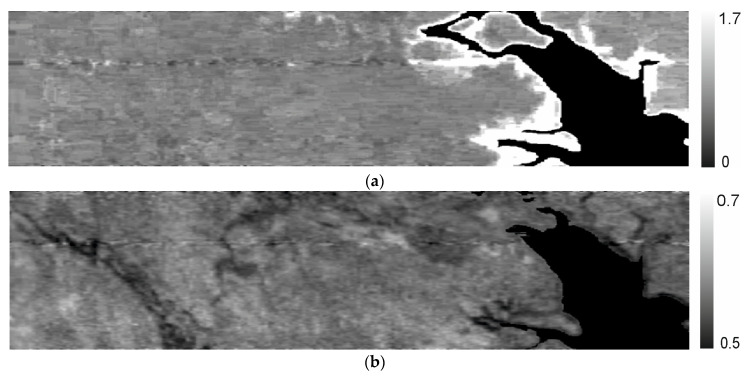
Distributions of the (**a**) water index (WI) and (**b**) anthocyanin reflectance index (ARI) of vegetation.

**Figure 19 sensors-24-02885-f019:**
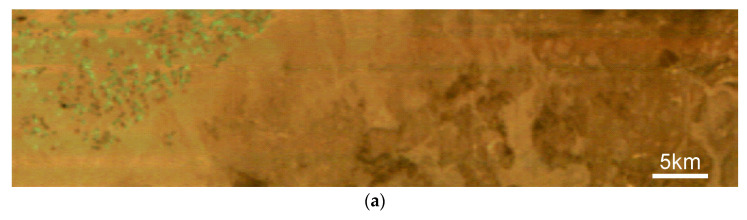
(**a**) Color-composite image generated from the hyperspectral data of the SXC3-219 (ISOI) satellite, (**b**) corresponding color-composite image from Sentinel-2 L2A, and (**c**) NDVI distribution calculated using the hyperspectral image from SXC3-219 (ISOI).

**Figure 20 sensors-24-02885-f020:**
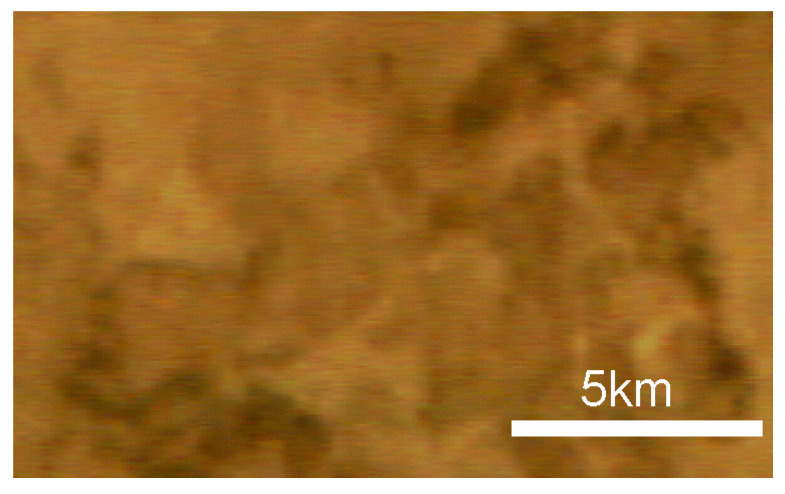
Magnified fragment of the image in Figure 19a.

**Table 1 sensors-24-02885-t001:** Relative response of the Sony IMX296 sensor across various wavelengths as specified by the manufacturer.

Wavelength, nm	Relative Response	Wavelength, nm	Relative Response
400	0.5	725	0.8
425	0.635	750	0.73
450	0.73	775	0.67
475	0.815	800	0.58
500	0.85	825	0.46
525	0.9	850	0.42
550	0.945	875	0.34
575	0.99	900	0.28
600	1	925	0.2
625	0.98	950	0.17
650	0.96	975	0.11
675	0.92	1000	0.08
700	0.87		

**Table 2 sensors-24-02885-t002:** NDVI values derived from ISOI and Sentinel-2 L2A data.

Point Number	NDVI (Sentinel-2 L2A)	NDVI (ISOI)
1	0.68	0.59
2	0.54	0.61
3	0.88	0.77

## Data Availability

Data underlying the results presented in this paper are not publicly available at this time but may be obtained from the authors upon reasonable request.

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
