# Peer review of "3U CubeSat-Based Hyperspectral Remote Sensing by Offner Imaging Hyperspectrometer with Radially-Fastened Primary Elements"

_sensors, 2024, doi:10.3390/s24092885_

Round 1
Reviewer 1 Report
Comments and Suggestions for Authors
This paper presents findings from a spaceborne Earth observation experiment utilizing a 12 novel, ultra-compact hyperspectral imaging camera aboard a 3U CubeSat. The performance of imaging high spectrometer is evaluated by numerical simulation. The analysis of hyperspectral data shows the applicability of this method in visible light and near infrared hyperspectral imaging. Compared with other imaging cameras, it has the potential to promote remote sensing applications. This paper has a certain application prospect, but there are also many problems that need to be corrected:
1. in figure 1, I think you should label each structure individually.
2. Are figure3 and 4 truncated? It is recommended to replace it with your own diagram to make it clearer
3. For the formula, it is recommended to specify each letter.
4. Figure 8, Figure 9 and Figure 10 have the problem of missing horizontal and vertical coordinates, so it is suggested to correct it.
5. Is Figure 12 definitely drawn from data? Please confirm it.
6. The pictures of the whole article are highly irregular, please confirm them again to ensure that they are clear and intuitive
Reviewer 2 Report
Comments and Suggestions for Authors
While I enjoyed and was greatly informed about cubesats and their applicability, especially the hyperspectral application shown in your manuscript, I do have a few suggestions:
1) Introduction: please describe the differences between all the various cubesat formats. Also there are several instances of the incorrect use of hyphens throughout this portion of the manscript (e.g., page 2 line 49 "satel-lites".
2) Section 4 Space experiment: There are numerous instances throughout this section in which the Sentinel satellite has been misspelled. Also, Equation 11 describes the bands used to calculate NDVI values, are these Sentinel 2 bands or cubesat bands? Further, which wavelengths do these correlate with?
3) Section 4 Figure 16: Please show on each image the general range of NDVI values in the SXC3-219 image and in the Sentinel image. This will help in comparing the images and will support the location-specific NDVI values in Table 2. Finally, what are the image dates and times of these images?
Comments on the Quality of English LanguageThe quality of the English use in the manuscript is excellent.
Reviewer 3 Report
Comments and Suggestions for Authors
General comments:
The authors have developed a low-cost and compact hyperspectral camera for remote sensing and put it into orbit.
Due to the spatial resolution and data rate, it is essential to define precisely the scope of use for this spectrometer satellite.
The manuscript must be written in the third person.
Specific comments:
Line 2. The manuscript title is too general.
Line 18. The word 'impressive' is unsuitable in this context.
Lines 18 and 80. “12-nanometer spectral resolution” and “5 nm spectral resolution”. Check the spectral resolution of your hyperspectral camera.
Line 27. Keywords should not repeat terms from the title of the manuscript, e.g. “hyperspectral remote sensing”. Few keywords.
Fig. 1. The blocks in the figure need to be numbered.
Fig. 3, 5, 8, 9... Axes must be labelled in all figures.
Tabl. 1. In the table and throughout the text, commas should be replaced with full stops.
Fig. 12. The figure captions are not legible.
Line 380. Considering the study's objective, it may be appropriate to omit Section “Ground testing of the hyperspectrometer”.
Line 389. The title is inappropriate.
Fig. 14. Please provide the scale, coordinates, and north orientation for the map. Additionally, if the picture was not taken with your camera, please ensure proper citation.
Line 426. Please provide a clear definition of B7 and B4 and include the formula used to calculate NDVI from the satellite data SXC3-219 (ISOI).
Fig. 16. Provide the NDVI scale for each image.
Fig. 17, Tabl. 2. Please provide the dates for SXC3-219 (ISOI) and Sentinel-2 L2A.
Lines 442-448. Statistical methods should be used to compare results. It is important to note that drawing conclusions based on only three points may not be appropriate.
Fig. 18, 19, 20. A scale must be introduced.
Round 2
Reviewer 1 Report
Comments and Suggestions for Authors
Thank you for your revision, which is more standardized than the first overall article, but there are some problems that need to be corrected:
1. When quoting relevant literature In line 58, relevant serial numbers should be consistent as on Line 52[11-14] and should not be listed separately.
2. In line 149 "Oprical transfer function(OTF)" has been abbreviated in the previous article, if used in the article, it should be abbreviated.
3. Indented error In line 164.
4. Formulas 11 and 12 need to be confirmed.
5. The format of references is wrong in line 521, and the format of all references needs to be unified.
